# Three-Dimensional Craniofacial Landmark Detection in Series of CT Slices Using Multi-Phased Regression Networks

**DOI:** 10.3390/diagnostics13111930

**Published:** 2023-06-01

**Authors:** Soh Nishimoto, Takuya Saito, Hisako Ishise, Toshihiro Fujiwara, Kenichiro Kawai, Masao Kakibuchi

**Affiliations:** Department of Plastic Surgery, Hyogo Medical University, Nishinomiya 663-8501, Japan; kane_and_peace@yahoo.co.jp (T.S.); hisako@hyo-med.ac.jp (H.I.); fuji-t@hyo-med.ac.jp (T.F.); k-kawai@hyo-med.ac.jp (K.K.); mkaki@hyo-med.ac.jp (M.K.)

**Keywords:** multi-phased deep learning, regression neural network, coordinate value, computer-assisted tomography (CT), craniofacial bone

## Abstract

Geometrical assessments of human skulls have been conducted based on anatomical landmarks. If developed, the automatic detection of these landmarks will yield both medical and anthropological benefits. In this study, an automated system with multi-phased deep learning networks was developed to predict the three-dimensional coordinate values of craniofacial landmarks. Computed tomography images of the craniofacial area were obtained from a publicly available database. They were digitally reconstructed into three-dimensional objects. Sixteen anatomical landmarks were plotted on each of the objects, and their coordinate values were recorded. Three-phased regression deep learning networks were trained using ninety training datasets. For the evaluation, 30 testing datasets were employed. The 3D error for the first phase, which tested 30 data, was 11.60 px on average (1 px = 500/512 mm). For the second phase, it was significantly improved to 4.66 px. For the third phase, it was further significantly reduced to 2.88. This was comparable to the gaps between the landmarks, as plotted by two experienced practitioners. Our proposed method of multi-phased prediction, which conducts coarse detection first and narrows down the detection area, may be a possible solution to prediction problems, taking into account the physical limitations of memory and computation.

## 1. Introduction

Measuring the distances between characteristic landmarks and the angles between certain planes determined by points is a useful approach to determining the shape of an object. This approach has long been used to evaluate human skulls [1,2,3,4]. However, it is impossible to obtain direct access to these landmarks when measuring the skulls of living humans. X-ray imaging makes it possible to project the skull. In the 1920s, Todd and Broadbent developed a device capable of holding human skulls and mandibles, which allowed for the acquisition of standardized radiographs [5]. Cephalometry, which was first introduced by Broadbent [6] and Hofrath [7] in 1931, remains one of the most helpful modalities for evaluating cranio-maxillo-facial configurations. Geometrical assessments are performed based on anatomical landmarks [8,9,10,11]. By utilizing cephalometry, clinicians and researchers can identify important characteristics and anomalies of the craniofacial region. The information forms the basis of surgical planning and future research. Various parameters based on cephalometry can be used to establish baselines and track changes over time.

Locating anatomical landmarks requires time and expertise. The automatic detection of such landmarks would provide significant medical and anthropological benefits. A number of studies have been conducted with the aim of accomplishing this challenge in 2D cephalograms [12,13,14,15,16,17,18,19,20,21], and a systematic review was published [22] in 2008. However, it was not possible to compare the studies’ methods, as they all used their own data, so were different from one another. The unification of the image data was necessary in order to carry out an assessment. Significant challenges were presented in 2014 [23] and 2015 [24], in conjunction with the IEEE International Symposiums on Biomedical Imaging. Cephalometric X-ray images with the coordinate values of landmarks were provided by the organizers. Participants competed by applying their own approaches to the same datasets. The top positions were achieved by contestants who used the random forest method [24]. Lindner et al. [25] used the same images with a subset of coordinate values. After that, a method using convolutional neural networks was proposed [26]. It outperformed the previous benchmarks with the same datasets used in the challenge. The use of multi-phased regression deep learning neural networks [27] with regression voting [28,29] enhanced the prediction accuracy. Kim et al. utilized a two-stage method with their own larger datasets [30]. Multi-staged convolutional neural networks were used on two-dimensionally projected cone beam computed tomography (CBCT) [31]. A study with an attentive feature pyramid fusion module [32] has surpassed previously published works. A systematic review of the use of artificial intelligence in cephalometric landmark identification was recently conducted [33]. A new challenge with larger datasets is currently underway [34].

The evaluation of objects as two-dimensionally projected images necessarily entails some inaccuracies and failures. Some of the original information contained in three-dimensional forms is lost or obscured by reducing the expressive dimension. It is rare for the cranium to be bilaterally symmetrical. There is often a significant degree of asymmetry between the left and right sides. In observing lateral cephalograms, it is common for the positions of the left and right mandibular angles to be misaligned in the images.

Three-dimensional cephalometric analysis was originally conducted with two (lateral and basilar or posteroanterior) cephalograms [35,36,37,38]. Computer-assisted tomography (CT) has become popular in daily clinical practice. Horizontal slices are presented as two-dimensional pictures, and CT images are usually stored in DICOM (Digital Imaging and Communications in Medicine)-formatted files. They can be digitally restructured into virtual three-dimensional objects. Additionally, 3D printing can be carried out, which enables people to visually comprehend the objects being studied. Three-dimensional measurements can be performed on the objects based on the anatomical landmarks. The cephalometric analysis of the three-dimensional images is becoming increasingly popular [39,40,41].

In comparison with the reports on 2D cephalograms, reports on the automatic landmark detection systems for 3D images are relatively new and fewer in number [42]. Shahidi et al. [43] used an atlas-based method to identify 14 landmarks from 20 CBCT images in 2014. A knowledge-based method [44] was reported in 2015. Various kinds of learning-based methods [45,46,47,48,49,50,51,52,53,54,55] have been reported. In our experience with 2D cephalograms [27,28,29], a multi-phased deep learning system was able to predict coordinate values with high precision. It first made a rough prediction for the whole area of the image; then, it marked down smaller areas of interest in the following phases. In this report, we present a multi-phased deep learning system for predicting the three-dimensional coordinate values of craniofacial landmarks in sequences of CT slices.

## 2. Materials and Methods

### 2.1. Personal Computer

All procedures were performed on a desktop personal computer: CPU (central processing unit)—AMD Ryzen 7 2700X 3.70 GHz (Advanced Micro Systems, Sunnyvale, CA, USA), memory—64.0 GB, GPU—GeForce RTX2080 8.0 GB (nVIDIA, Santa Clara, CA, USA), and Windows 10 pro (Microsoft Corporations, Redmond, WA, USA). Python 3.7 (Python Software Foundation, Wilmington, DE, USA), a programing language, was used under the Anaconda 15 (FedoraProject. http://fedoraproject.org/wiki/Anaconda#Anaconda_Team_Emeritus, accessed on 30 May 2023) as an installing system, and Spyder 4.1.4 was used as an integrated development environment. Keras 2.31 (https://keras.io/, accessed on 20 February 2020), a deep learning library written in Python, was run on TensorFlow 1.14.0 (Google, Mountain View, CA, USA). The reason for using Keras with Tensorflow was that there are active communities of developers and researchers for them, providing vast varieties of pre-developed or pre-trained models. GPU computation was employed through CUDA 10.0 (nVIDIA). For 3D reconstructions, slicer 4.11 (www.slicer.org, accessed on 20 February 2021) was used with the Jupyter Notebook (https://jupyter.org/, accessed on 20 February 2021). OpenCV 3.1.0 libraries (https://docs.opencv.org/3.1.0/, accessed on 20 February 2021) were used in the image processing.

### 2.2. Datasets

#### 2.2.1. CT Images

A collection of CT images from head and neck squamous cell carcinoma patients was retrieved from the public access Cancer Imaging Archive (wiki.cancerimagingarchive.net) Head-Neck-Radiomics-HN1 dataset [56]. The dataset consists of a folder for each patient, containing 512 × 512 px DICOM images, taken axially at 5 mm intervals in the cephalocaudal direction: 512 px correspond to 500 mm, as the pixel spacing recorded in DICOM tag was (0.9765625, 0.9765625). The order of the images was checked and images from the top of the head to the mandible were extracted for 120 cases. The images from the caudal to the area of interest were excluded. The largest number of images extracted for a single patient was 81. The direct reads of the numbers in the images ranged from 0 to 4071. To standardize the coordinate values and their origins, the size of all the reconstructed 3D objects was set to 512 by 512 px in the x and y directions. For convenience, in the subsequent procedures, pixels were used as the unit of measurement. As a calibration marker, a cross of 512 px in length and width was added to the most caudal image of each patient. Briefly, from each patient folder, the last DICOM file was selected. Using Pydicom, a Python module that deals with DICOM files, the image was read from the file. Using NumPy, another Python module used for mathematical operations, all elements with an index of 255 in the 0 and 1 dimensions were replaced by the number 4000. Using Pydicom, the original image in the DICOM file was replaced by the image with the calibration cross and was then saved.

#### 2.2.2. 3D Reconstruction (STL File Creation)

The DICOM CT image sequence for each case was processed with a 3D slicer kernel using Jupyter Notebooks. Using a python script process [57], sequences of DICOM files for patients were loaded. Bony parts were segmented into groups of 250 and reconstructed into 3D objects. The size of the objects, including the blank areas, was 512 × 512 along the x- and y-axes and varied along the z-axis, depending on the patient. They were stored as STL files. The script produced STL objects for all patients in a series. It took 1 h and 40 min to convert all 120 of the DICOM file series into STL images.

#### 2.2.3. Plotting Anatomical Landmarks

Each STL file was imported into blender (https://www.blender.org/, accessed on 20 February 2021). For each anatomical landmark, a colored sphere with a radius of 1 px was placed in the corresponding location. Five perspective views of the object were displayed on a screen to ensure the location of the spheres in the x, y, and z directions. The objects with the spheres were saved in the STL format. To maintain consistency, the spheres for a given landmark were placed on all the objects in the series. After one landmark had been plotted for all the objects, another landmark plotting procedure was undertaken. The plotted landmarks are listed in Table 1 and shown in Figure 1. Most of the images were likely from elderly patients, as there were many missing teeth. Many of the images were taken in the open bite position. Therefore, landmarks on teeth were not plotted in this study. Using Python scripts for blender, the three-dimensional coordinate values (x, y, z) of the spheres on the STL objects were obtained. They were exported as an array of 120 cases × 16 points × 3 in the csv format. Two practitioners, with 31 and 10 years experience, plotted the landmarks. The coordinate values plotted by the senior practitioner were used as the ground truth.

### 2.3. Neural Networks and Learning Datasets

Ninety cases were designated as training data and thirty were used as testing data.

#### 2.3.1. First-Phase Deep Learning

Each CT image in the DICOM files in the folders of the cases used for training was read using Pydicom (512 × 512 px). Using OpenCV, the CT image was compressed to 96 × 96 px. It was binarized to segment bone with 1100 as the threshold and then converted to a 0 or 1. Arrays of zeros with a 96 × 96 × 81 shape were prepared for a three-dimensional template. For each case, compressed images replaced the template from the bottom and were stacked up. A modified regression deep learning model (only the last activation layer was changed, from “softmax” to “linear”), Resnet 3d-50 [58], was built, with 96 × 96 × 81 as the input and 48 as the output. The coordinate values of 16 landmarks were set as the targets, the batch size was set to 8, and it was trained for 150 epochs. The weights of the trained model were saved as a file (Figure 2).

#### 2.3.2. Second-Phase Deep Learning

Each CT image in the DICOM file was read with Pydicom. A 100 × 100-px image was cropped out from each original image using OpenCV, centered on the x and y coordinates of each landmark. It was binarized to a 0 or 1 using the threshold of 1100. All the cropped and binarized images were stacked up from the bottom to form a 100 × 100 × 81 3D array. The target coordinate values were 50 for x, 50 for y, and the ground truth z value for z. For data augmentation, the images were also cropped out at shifted positions in the x and y directions, from −30 to 30, in 10-px steps. They were stacked in the same way to obtain the positions of the feature points in each array (3240 sets in total). The target coordinate values along the x- and y-axes for these shifted images were changed with the amount of shift. For each landmark, the modified Resnet 3d-50 model was prepared for regression with 100 × 100 × 81 as the input and 3 as the output. In total, 16 models with a batch size of 16 were trained for 100 epochs. The weights of the trained models were saved (Figure 3).

#### 2.3.3. Third-Phase Deep Learning

A 50 × 50-px image was cropped out from each original image, using the same method described for the second phase. Stacks of 50 × 50 × 81 were obtained. The target value for x was 25, for y it was 25, and for z it was the ground truth z value. For data augmentation, the images were also cropped out at shifted positions in the x and y directions, from −15 to 15 in 5-px steps. The target values for x and y were changed as the range shifted. For training, 3240 sets of data for each landmark were used. A modified Resnet 3d-50 model with an input of 50 × 50 × 81 and an output of 3 was constructed. In total, 16 models with a batch size of 16 were trained for 150 epochs. The trained weights were saved (Figure 4).

### 2.4. Evaluation

For the evaluation, 30 testing datasets that were not used during training were employed (Figure 5). It took 52.5 s to load the modules and 33 models. Landmark predictions for 30 patients using 3-phase models took 33 min and 3.0 s.

#### 2.4.1. First-Phase Prediction

The first-phase model and the trained weights were loaded. The 96 × 96 × 81 3D-arrays of the 30 testing cases were fed to the model to predict the 3D coordinates of the feature points. They were saved in a csv file. The prediction errors for the distances between the predicted point and the ground truth landmark were calculated in x, y, and z. The three-dimensional prediction errors were presented as the square root of the sum of the squares of the gaps along the three axes.

#### 2.4.2. Second-Phase Prediction

The 100 × 100-px images were cropped from the original validation images and centered on each of the 16 coordinates obtained during the first-phase prediction. The coordinate values for the starting point of the cropped object in the original objects were recorded. The cropped images were piled up into 100 × 100 × 81 3D arrays. They were used to predict the coordinates of each feature point in the cropped object with the trained second-phase models for the respective landmarks. The predicted coordinate values of the 16 points were saved.

#### 2.4.3. Third-Phase Prediction

For each landmark, 50 × 50-px images were cropped, centering on each of the coordinates obtained in the second-phase prediction. The coordinate values for the starting point of the cropped object in the original objects were recorded. They were stacked up into 50 × 50 × 81 arrays and fed to the respective 16 trained third-phase models. The predicted values were recorded.

#### 2.4.4. Prediction Error Evaluation

The distance norm between the predicted coordinates and the manually plotted ground truth coordinates was calculated as the absolute value in the x, y, and z directions. The square root of the sum of the squares of each was used as the 3D distance. For the second-phase and third-phase predictions, the coordinate values of the predicted points in the original objects were configured with the starting point for the cropped object in the original object and the predicted coordinate in the cropped object. The prediction errors were calculated as the distances between the predicted points and the ground truth landmark points in the original object.

#### 2.4.5. Statistical Analysis

Multiple comparisons were conducted using scikit-posthocs (https://scikit-posthocs.readthedocs.io/en/latest/#, accessed on 20 February 2021).

## 3. Results

### 3.1. First-Phase Prediction Error

Overall, the average three-dimensional distance between the predicted points and the ground truth was 11.60 px (1 px = 500/512 mm) (Table 2). The per-landmark prediction errors are shown in Figure 6. Regarding the axis directions, the error for the x-axis was significantly smaller than that for the others, and the error for the y-axis was the largest (Figure 7).

### 3.2. Second-Phase Prediction Error

The average prediction error in three dimensions was 4.66 px (Table 2). It was significantly smaller than the error for the first-phase prediction. The per-landmark errors are shown in Figure 8. The error for the y-axis direction was larger than that for the other directions, and the error for the z-axis was the smallest (Figure 9).

### 3.3. Third-Phase Prediction Error

The three-dimensional prediction error was 2.88 px, on average (Table 2). It was significantly smaller than the second-phase prediction error. The errors per landmark are shown in Figure 10. There were no significant differences between the axes (Figure 11).

### 3.4. Inter-Phase and Inter-Observer Plotting Gaps

There were significant differences in the prediction errors for each phase. The third-phase prediction error was the same level as the inter-observer plotting gaps (Table 2).

In summary, three-dimensional prediction errors became smaller as the phase progressed. The errors in the 3rd phase prediction showed no significant differences in the inter-observer gaps between the two experienced practitioners. By the 1st and 2nd phase prediction, the errors in the y-axis direction were larger than the x- and z-axis directions. By the 3rd phase, the differences dwindled. For each landmark, there was some variation in prediction accuracy in the 1st and 2nd phase but no significant variation in the 3rd phase.

## 4. Discussion

Since it was first proposed by Broadbent [6] and Hofrath [7], standard cephalometry has had a rich history. There have been many reports on the use of automatic landmark detection systems for cephalograms in two dimensions. In comparison with the research into 2D cephalograms, there are few reports on 3D images.

Previous automated 3D landmark detectors for use in the craniofacial area have employed the registration method [59], knowledge-based methods [44,60], atlas-based methods [43,61], random forest methods [45], and so on [47,49]. Deep learning is an emerging technique in machine learning; it has attracted increased attention in recent years due to its remarkable ability to learn from vast amounts of data and make accurate predictions. It is categorized as a supervised learning method that identifies rules between the input and output of training datasets. It is simply to prepare the datasets, and the machine will then determine the function laws between them. Due to its versatility, deep learning is widely used in a variety of fields, including natural language processing, computer vision, speech recognition, robotics, voice, and text. Some attempts to utilize deep learning for automatic 3D landmark prediction have been reported [46,50,51,52,53,54,55].

The processing speed of computers and the amount of memory installed have increased at a remarkable rate. These technological advancements have revolutionized various fields, including image processing, which is one of the fields where deep learning algorithms have become increasingly popular. However, an enormous amount of computation is still required to process images using deep learning. The calculation volume required to process three-dimensional images, based on spatial or time axes, is on a completely different scale than that used to process their two-dimensional counterparts. There are two reports of 3D landmark detection using fully convolutional neural networks (FCN) with high precision [49,58]. However, in general, FCN is computationally expensive and slow [62].

One solution is to compress the images [51] and then input them into deep learning networks, but the compression process results in the loss of detailed information. In this study, we took the multi-phase deep learning method, which is used to predict landmarks in 2D cephalograms [27,28,29], and applied it to 3D craniofacial images. The conceptual premise was to emulate the way that one finds a place on a map when provided with the address. First, we open a map of the country. Then, we try to find the state and city on a larger-scale map. Then, we open an even larger map to find the street and house number. Prediction errors became smaller with each phase. Coarse detection was performed with the first phase model, and further narrowing down was achieved based on the predictions of the previous phase. This study was conducted on a personal computer. Accounting for the physical limitations of memory and computation, multi-phase deep learning may be a feasible means of dealing with large-scale images. This coarse-to-fine detection concept has been adopted by other authors. Yun et al. [55] utilized a variational autoencoder to achieve a rough initial estimate. They are unique in differentiating fine detection methods for the mandible and cranium. For the mandible, they achieved an estimation error relative to the reference of 2.68 mm; this value was 3.08 mm for the cranium and 2.88 mm for the 90 total landmarks. Dot et al. [54] trained a spatial configuration network [62] on down-sampled-resolution full scans. For the fine resolution within a selected region of interest, they trained five spatial configuration networks. Using the 2-stage method, their mean localization error was as fine as 1.0 mm for 33 landmarks.

Logically speaking, our system is very simple. Through all three phases, the main part of the models used was the same Resnet 3d-50, modified for regression. However, the system consists of 33 networks that are individually trained to predict 16 landmarks, and it cannot be denied that it is structurally complicated. There may be ways to design the system in an end-to-end fashion. Again, given the calculation limit, this sequential approach was practical for the authors and remains a promising option for future research in this area.

Previous studies [27,28,29] of 2D cephalograms used the database that was published at ISBI 2015, along with previous benchmarks [24,26]. The authors were unable to obtain a database of feature-point three-dimensional coordinates for craniofacial CT. The authors of the previous reports [43,44,45,46,47,48,49,50,51,52,53,54,55] on the three-dimensional landmark detection used their own data, which are not publicly accessible. Since the previous studies all use different datasets, their prediction accuracy could not be compared with the other published methods. Unified datasets of 3D images and landmark coordinate values should be published, as they were for the 2D challenges [23,24,34]. Generally, in studies using data with low variance, the prediction error is small; meanwhile, in studies that use highly dispersed data, the prediction error is large. This is because the greater the variance of the data, the greater the uncertainty in the underlying relationships and patterns, making it more difficult to predict accurately. The size of the data volume can also have a significant impact on the prediction error. In general, higher data volumes can help to reduce prediction errors by providing more comprehensive and representative samples that contain information about the underlying phenomenon. The higher the data volume, the more uncertainties tend to be eliminated. Systems based on high-volume datasets tend to have high levels of robustness.

As for the current situation in Japan, it is not easy to access and build databases of patient information, even for clinicians, such as the authors of this article. The same point has been made by Yun et al. [55]. Hence, the authors constructed an original database from publicly available image sets [56]. The image sets used were from patients with head and neck tumors. Most of the images were likely from elderly patients, and there were many missing teeth.

In the course of conducting our study, we encountered a significant challenge in plotting anatomical landmarks. We adopted blender in this procedure as we were familiar with it. Plotting the featured points one by one took a long time. Moreover, in order to capture a comprehensive three-dimensional image of each feature point, multiple perspective views were necessary. In this study, we used five viewpoints in conjunction. In addition, it is often necessary to pan and zoom in and out; bone ridges are formed by curves, not by sharp angles, so it was difficult and sometimes impossible to plot them accurately. To maintain as much consistency as possible, the series included in this study was produced by one person, and one feature point was plotted for all cases in succession. This method was used based on the belief that the continuous determination of points would produce high-quality data.

The craniofacial CT data used in this study were provided at 5 mm intervals in the cranio-occipital direction. The data intensity along the z-axis was more than five times sparser than that along the x or y-axes. However, to our surprise, prediction errors in the z-axis direction were not worse than for the x or y-axes (Figure 7, Figure 9 and Figure 11).

The first- and second-phase prediction errors in the y-axis direction were larger than those in the x- and z-axis directions. Looking back at the original CT images, the positions of the patients’ heads in the images were relatively centered in the x-axis direction, whereas they were spatially scattered in the y-axis direction. The degree of dispersion of the data may have led to estimation errors. The second-phase prediction was possibly affected by the first-phase prediction, as it was performed on the cropped images, which were defined by the first-phase prediction, whereas the y-axis prediction errors were at the same level as those of the other axes in the third phase. The impact of the first-phase prediction errors on the y-axis may have been diminished.

The third-phase prediction error revealed no significant differences from that of the two experienced practitioners. The system was ultimately performed at the clinician level. The issue that the plotted ground truth coordinate values may differ from one plotter to another has been a matter of debate [63]. There is no consensus on how precisely these systems should function. It is well known and should be taken as a given that inter-observer [63] and intra-observer errors always exist [64,65]. It seems reasonable to set goals at the inter-expert error level, and our system was able to attain that benchmark.

Automatic craniofacial landmark detection will contribute to sparing time and labor for various researchers. Accurate identification and localization of craniofacial landmarks are crucial for surgical planning in various craniofacial procedures. To understand the surgical impact on the patient, it is very important to monitor the positions and relationships of these landmarks over time. All practitioners, who perform surgeries on the craniofacial area, including plastic surgeons, maxillofacial surgeons, otolaryngologists, neurosurgeons, pediatric surgeons, orthodontists, ophthalmologists, and so on, will benefit from this kind of software. Forensic experts and anthropologists can save time comprehending the human head three-dimensionally.

In the field of clinical practice, slices of less than 1 mm are commonly used to obtain detailed bone information (so-called thin slices). To apply these images in real-world clinical situations, such as in the navigation systems utilized during surgery, it is necessary to have adequate support for these images. Our system, based on 5 mm thickness slices, cannot be directly applied to those thin-sliced detailed data.

CBCT has gained significant popularity among orthodontists and otolaryngologists; it has the benefit of exposing patients to less radiation than traditional CT scans, and it can obtain detailed images with small voxels. This has enabled highly accurate estimations and diagnosis of various dental and craniofacial conditions. However, as the amount of information to be processed increases, the computational volume required also increases. Our proposed method of multi-phased prediction, which begins with coarse detection and then narrows down the detection area, may be a possible solution to this problem.

## 5. Conclusions

A multi-phased deep learning system was constructed to predict landmarks on three-dimensional craniofacial images. It is a sequential system that first detects landmarks coarsely and marks down the region of interest in the following phases; it consists of 33 individual networks. The system reached the same level as expert clinicians. Given the limitations in computational resources, multi-phase deep learning may be a solution for dealing with large-scale images. There is a need for the publication of high-volume unified datasets of 3D images with landmark coordinate values; this would promote research in this area and enable the comparison of studies.

## Figures and Tables

**Figure 1 diagnostics-13-01930-f001:**
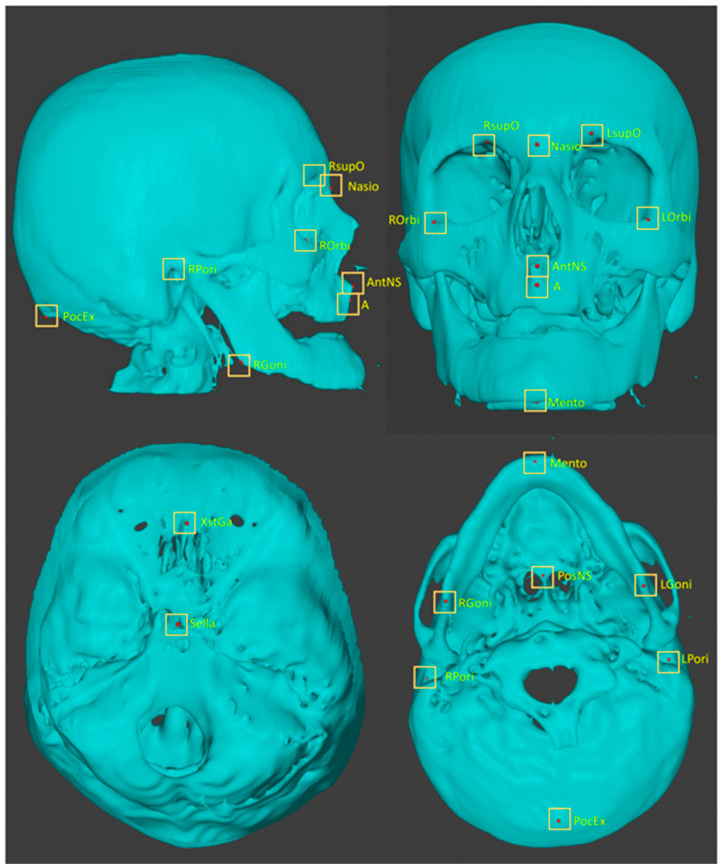
Three-dimensionally plotted landmarks.

**Figure 2 diagnostics-13-01930-f002:**
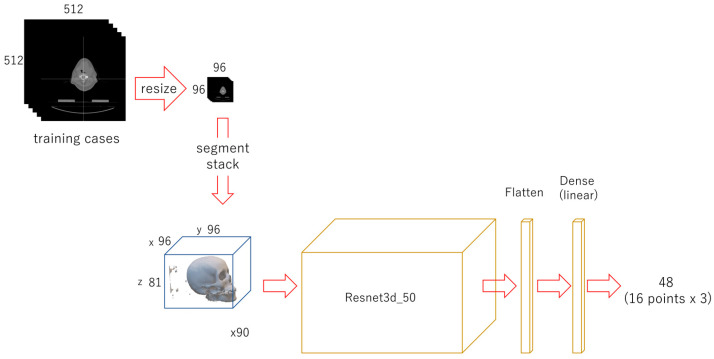
Diagram of the first-phase deep learning. Each CT image (512 × 512 px) was compressed to 96 × 96 px and stacked up. A regression deep learning model was trained with 90 cases for 150 epochs.

**Figure 3 diagnostics-13-01930-f003:**
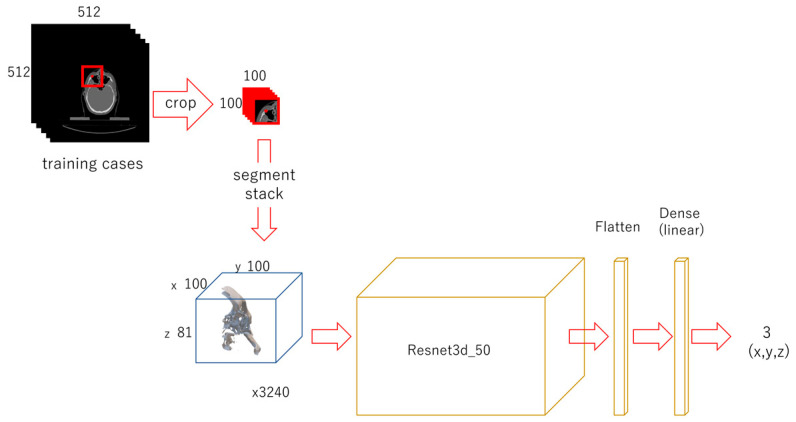
Diagram of the second-phase deep learning. For each landmark, 100 × 100-px images were cropped out and its x and y coordinates were centered. They were stacked up into a 100 × 100 × 81 3D array. The shifted images were also cropped and stacked for data augmentation. A model was trained for each landmark.

**Figure 4 diagnostics-13-01930-f004:**
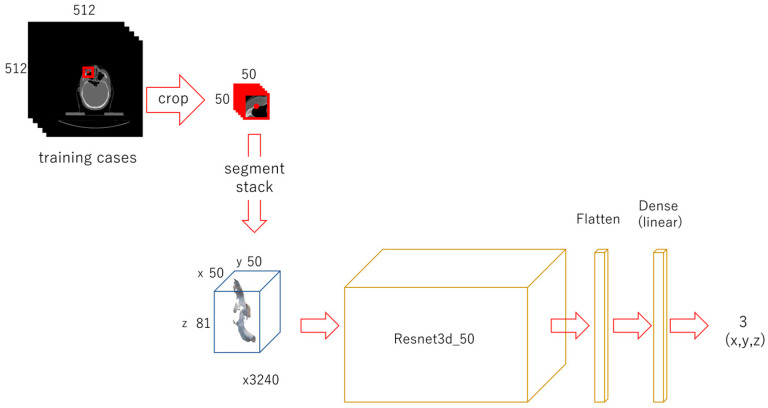
Diagram of the third-phase deep learning. A 50 × 50-px image was cropped from the original image, centering the x and y coordinates of the landmark. They were all stacked up to form a 3D array. The shifted images were also cropped and stacked. A Resnet model was trained for each landmark.

**Figure 5 diagnostics-13-01930-f005:**
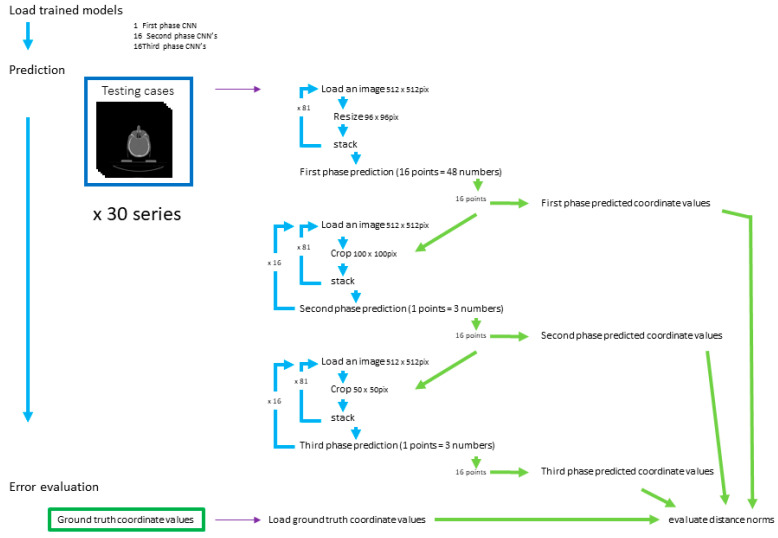
Prediction and evaluation. Prediction of the landmark coordinates was achieved in three phases. The three-dimensional distances between the predicted and ground truth points were evaluated.

**Figure 6 diagnostics-13-01930-f006:**
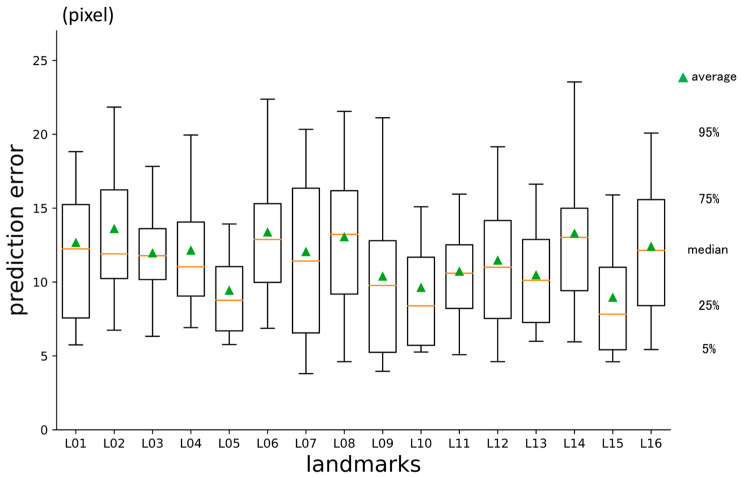
First-phase prediction errors per landmark (pixel = 500/512 mm).

**Figure 7 diagnostics-13-01930-f007:**
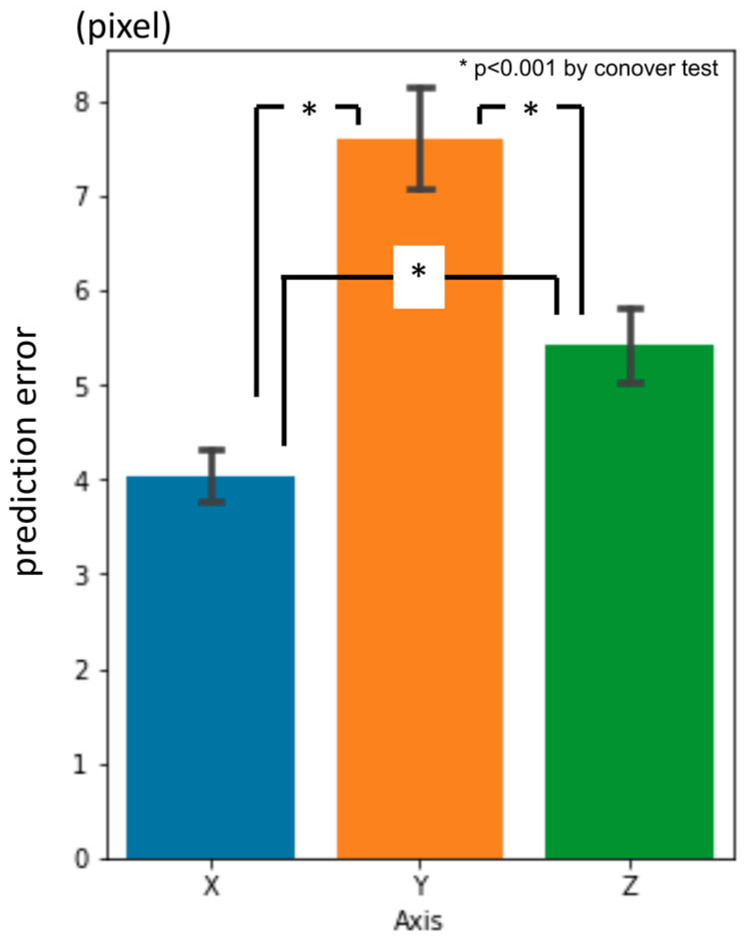
First-phase prediction errors per axis (pixel = 500/512 mm).

**Figure 8 diagnostics-13-01930-f008:**
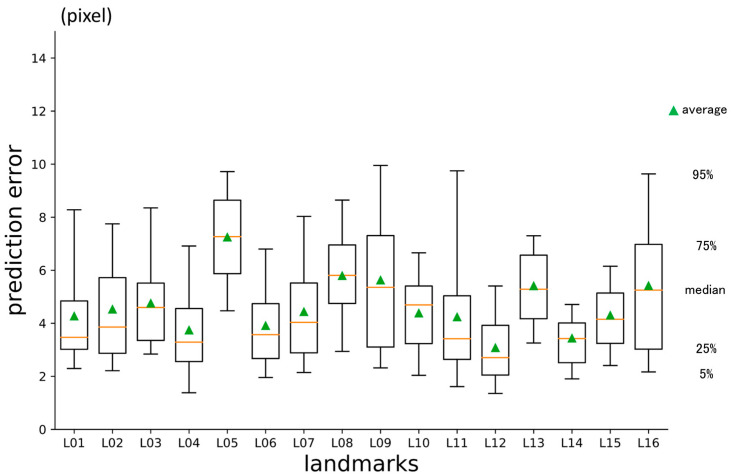
Second-phase prediction errors per landmark (pixel = 500/512 mm).

**Figure 9 diagnostics-13-01930-f009:**
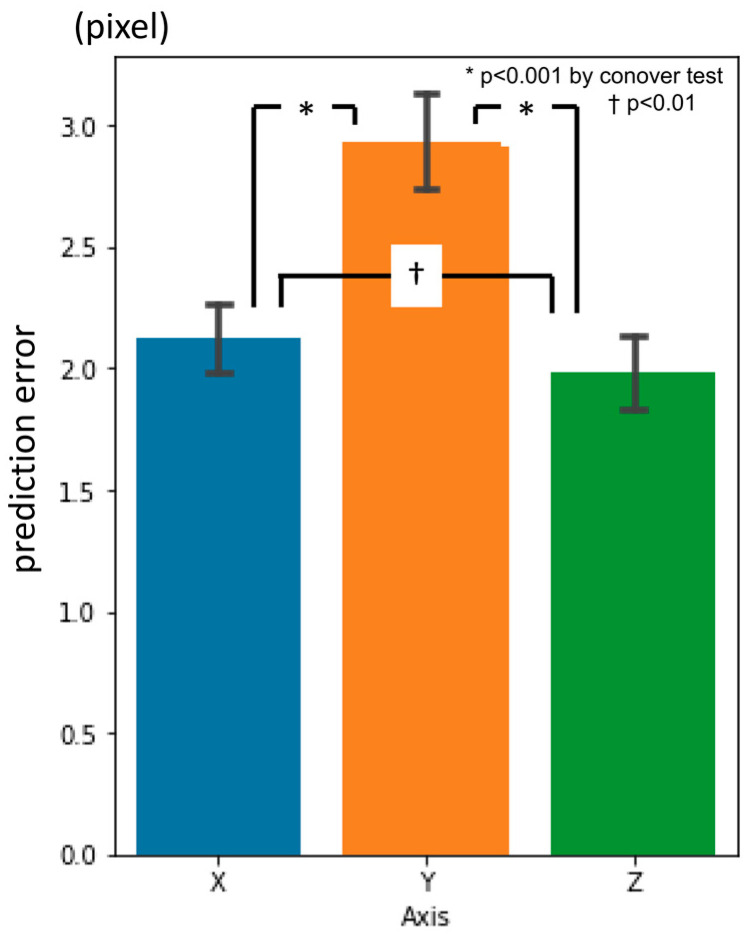
Second-phase prediction errors per axis (pixel = 500/512 mm).

**Figure 10 diagnostics-13-01930-f010:**
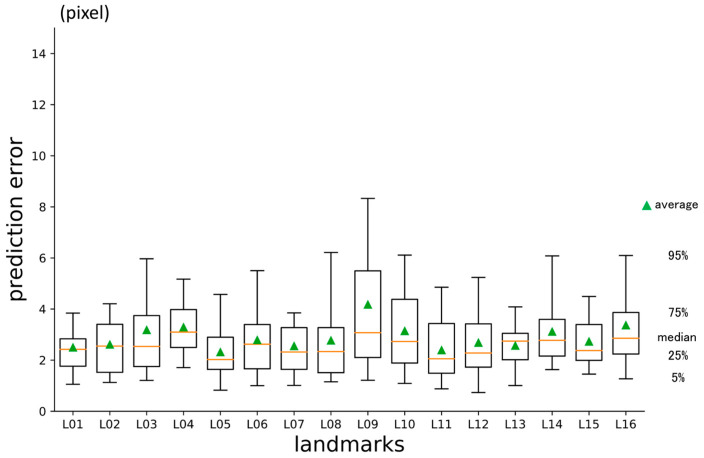
Third-phase prediction errors per landmark (pixel = 500/512 mm).

**Figure 11 diagnostics-13-01930-f011:**
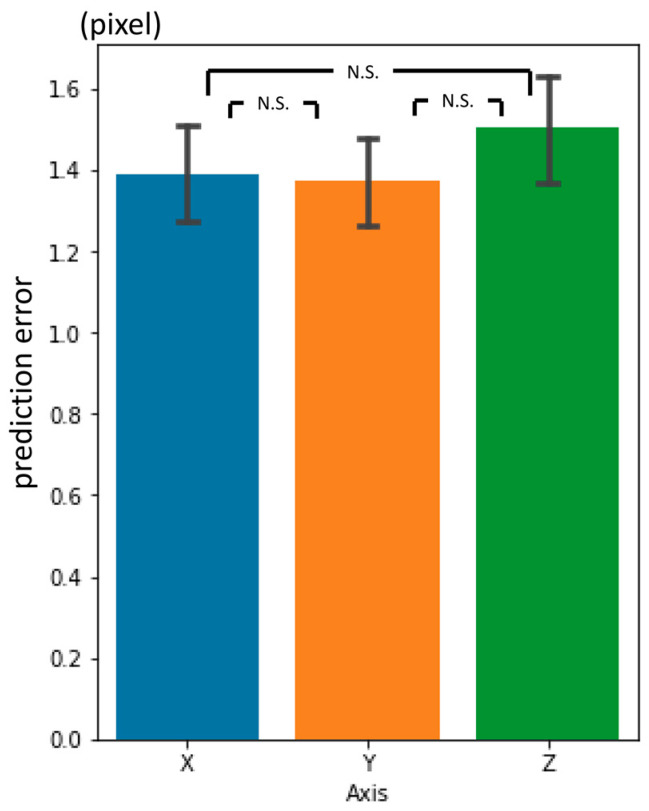
Third-phase prediction errors per axis (pixel = 500/512 mm). N.S.: not significant.

**Table 1 diagnostics-13-01930-t001:** Plotted landmarks.

No	Abbreviation	Description
L01	A	Point A
L02	AntNS	Anterior nasal spine
L03	LGoni	Left gonion
L04	LOrbi	Left inferior lateral orbital rim
L05	LPori	Left porion
L06	LsupO	Left supra orbital incisura
L07	Mento	Menton
L08	Nasio	Nasion
L09	PocEx	External occipital protuberance
L10	PosNS	Posterior nasal spine
L11	RGoni	Right gonion
L12	ROrbi	Right inferior lateral orbital rim
L13	RPori	Right porion
L14	RsupO	Right supra orbital incisura
L15	Sella	Center of sella turcica
L16	XstaG	Top of crista galli

**Table 2 diagnostics-13-01930-t002:** Three-dimensional prediction errors in 480 landmarks from 30 testing datasets. (pixel = 500/512 mm) * *p* < 0.001 in the Conover test. N.S.: not significant.

	First Phase		Second Phase		Third Phase		Inter-Observer Gaps
Average	11.6	_ * _	4.66	_ * _	2.88	_ N.S. _	3.08
Median	10.89		4.22		2.56		2.4
Stdev	5.64		2.27		1.67		2.64

## Data Availability

The source code of this manuscript is available at https://drive.google.com/drive/folders/1RI4ZHr2l2KyIpCZpV0KSLY2YVMbX0wAU?usp=sharing, accessed on 30 May 2023. Publicly available datasets were analyzed in this study. These data can be found here: https://wiki.cancerimagingarchive.net/display/Public/Head-Neck-Radiomics-HN1, accessed on 30 May 2023 [56].

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
