# Peer review of "Three-Dimensional Craniofacial Landmark Detection in Series of CT Slices Using Multi-Phased Regression Networks"

_diagnostics, 2023, doi:10.3390/diagnostics13111930_

Round 1

Reviewer 1 Report

In this manuscript, the authors used multi-phased deep learning networks to predict the three-dimensional coordinate values of the craniofacial landmarks. The authors used a publicly available database of computed tomography images of the craniofacial area to train and evaluate the deep learning networks. While the work is scientifically sound, the manuscript needs several clarifications and hence I suggest major revision as per the comments below.

1. The authors should provide more justification for their choices in processing steps. For example, why did they use Keras and Tensorflow, and not other deep learning libraries?

2. Line 108 in the methods section mentions “The number in the images ranged 0 to 4071” What is the number referring to? Is it grayscale intensity values? In the python code it seems that the grayscale values are being changed to 0-250, but this need more clarification in the methods section.

3.      It seems the authors used Slicer to threshold bone and then converted the segmentation to STL format for landmarking in Blender. Slicer also has an option to landmark 3D objects and it is not clear as to why the authors used an additional step to convert to STL and use Blender. This should be justified in methods or discussion section of the manuscript.

4.      Some of the sentences and descriptions are quite long and difficult to parse. For example, the description of the DICOM file processing in section 2.1 is quite convoluted and could be broken up into smaller, easier to understand sentences. Similarly, the description of the landmark plotting procedure in section 2.2 could be clearer and more concise.

5.      The results section can be expanded. It would be helpful to provide more context on what these prediction errors mean and how they compare to previous studies or benchmarks in the field. This would help readers better understand the significance of the reported errors. On the same note, the subheadings could be more descriptive to indicate what each section is about and what information is being presented. Additionally, it would be helpful to include a brief summary or conclusion at the end of the results section to summarize the key findings and what they mean for the study.

6.      The code provided in the Google Drive folder is not organized in a clear and structured manner. There are multiple scripts with different purposes, and it can be difficult to follow the flow of the code. The code lacks sufficient comments and documentation to explain what each section does and how it contributes to the overall objective. This can make it difficult for others to understand and modify the code.

Needs improvement of overall language in the manuscript as several sentences are hard to understand. 

Author Response

Thank you for reviewing our manuscript.

1: The reason for using Keras with Tensorflow was that there are active communities of developers and researchers for them, providing vast varieties of pre-developed or pre-trained models. Lines were added in 2.1 section.

2: The image data in DICOM format are saved as two-dimensional matrixes of pixel numbers. CT number (Hounsfield unit)  is a linear transformation of the original linear attenuation coefficient, defining radiointensity of water as 0 and that of air as -1000. They can reach up to more than 4000 for metal materials. In making CT images, windowing of the CT numbers are done to show appropriate appearance of the interested structures. So, the number range in the image data in DICOM varies. When converting the images to 8 bit formats (e.g. .png, .jpg and so on), the numbers are compressed into 0-255. Python can deal with images (matrixes of numbers), consisted of larger range. We do not think this information is needed for this paper. The line in 2.2 (1) was revised.

3: There are several reasons to adopt Blender. To plot the landmarks and obtain their coordinate values, 5 perspective views were displayed on a screen. It was easier for us to plot the same landmark sequentially on the series of cases in Blender. The main reason was that we are used to it. A sentence was added in Discussion.

4: We let the issues to the English editing services.

5: As was written in the manuscript, the prediction errors by the 3rd phase marked the same level between two experienced practitioners. It can be said that it is clinical level. It is worthless to compare the results with the studies done on the different datasets. That is why we stressed the importance of the publication of unified datasets of 3D images and landmark coordinate values. As was suggested, a brief summary of the results was added at the end of Results section.

6: I am sorry that the codes appear lousy. None of the authors are professional programmers. We are all surgeons. Those are the codes, we actually used for this study. So, rewriting the codes does not seem to be a good scientific attitude. Readers may realize that “this level” code writer was able to achieve the results, which may encourage them.

Reviewer 2 Report

I carefully read the manuscript „Three-dimensional craniofacial landmark detection in series of 2 CT slices, using multi-phased regression networks”. It is indeed interesting. I would’ve preferred to see also peculiar outputs of the method presented here presented as for practitioners: CT views with specific landmarks, if applicable. There should’ve been discussed how it could be specifically used by surgeons for example, and why? Orthodontists have CBCT and, for example Blue Sky Bio software, which is a complex and already existing program. The authors should have discussed why also ENT is in need for such craniofacial landmarks and when ENTs specifically determine these landmarks and are in need for a certain software. To me, this is a well done work without a certain future development. In these regards I keep doubts on the citabililty of this paper. I therefore ask the authors to specifically answer my queries by adding discussions to these comments of mine.

Minor English checks and corrections are needed.

Author Response

Thank you for reviewing our manuscript.

Automatic craniofacial landmark detection will contribute to sparing time and labor for various researchers. Accurate identification and localization of craniofacial landmarks are crucial for surgical planning in various craniofacial procedures. To understand the surgical impact on the patient, it is very important to monitor the positions and relationships of these landmarks over time. All practitioners, who perform surgeries on craniofacial area, including plastic surgeons, maxillofacial surgeons, otolaryngologists, neurosurgeons, pediatric surgeons, orthodontists, ophthalmologists, and so on, will have benefit from this kind of software. Forensic experts and anthropologists can save time in comprehending the human head three-dimensionally. Lines were added in Discussion.

Round 2

Reviewer 1 Report

In the Methods section 2.2 (1), line 108, you mention '512 x 512 pixel DICOM images, taken axially at 5 mm intervals in the cephalo-caudal direction: 512 pixels correspond to 500 mm.' Could you clarify how 512 pixels correspond to 500mm? 

The English language has been substantially revised. 

Author Response

Thank you for reviewing our manuscript.

That kind of information is recorded in DICOM tag (0028,0030): Pixel Spacing. In this series of images, it was (0.9765625, 0.9765625). The line was modified.

Reviewer 2 Report

The authors answered my demands.

Author Response

Thank you for reviewing our manuscript.